# A Multi-Criteria Approach to Assess the Performance of the Brazilian Unified Health System

**DOI:** 10.3390/ijerph191811478

**Published:** 2022-09-13

**Authors:** Renan Felinto de Farias Aires, Camila Cristina Rodrigues Salgado

**Affiliations:** 1Department of Applied Social Sciences, Federal Rural University of the Semi-Arid, Mossoró 59625-900, Brazil; 2Department of Applied Social Sciences, Federal University of Paraíba, Bananeiras 58051-900, Brazil

**Keywords:** health system, unified health system (SUS), index of unified health system performance (IDSUS), performance assessment, multi-criteria decision analysis (MCDA), R-TOPSIS

## Abstract

Brazil’s Unified Health System (SUS) provides universal free access to health services and is considered a model for the rest of the world. One of the tools used by the Brazilian government to assess this system is the Index of Unified Health System Performance (IDSUS). However, this method has a number of limitations, such as disregarding the opinion of healthcare decision makers. Thus, the aim of the present study was to propose a model based on the R-TOPSIS in order to assess the performance of the SUS. Methodologically, the main steps for proposing multi-criteria models were followed, and to validate the model, a real case study with a set of six cities (alternatives) of the state of São Paulo was used. The results provide a clearer picture of the differences in terms of potential and obtained access, as well as the effectiveness of health services in the cities analyzed. Likewise, the proposal of integrating multiple criteria as well as considering healthcare decision makers proved to be decisive for the results obtained, even in comparison with the other approaches. It was concluded that the proposed method provides a robust and adequate analysis of health systems performance.

## 1. Introduction

It is important to recognize the crucial role of public health in improving the health of society and reducing inequalities [1]; however, most of the decisions in this area are made under conditions of uncertainty [2]. In addition to uncertainty, decisions are complex and involve the preferences and values of stakeholders, which justifies the proposal of using several methods to help and support them [3].

In the specific case of decision making for public health systems, this complexity is even more marked. These systems are responsible for protecting the health and well-being of the communities they serve [4]. As such, any attempt to assess or characterize a high-performance public health system requires a thorough analysis [5].

A number of studies have assessed the performance of healthcare systems, such as those by Gramani [6], Gerring et al. [7], Vandan et al. [8], and Tille et al. [9]. The first of these involved an investigation of the performance of Brazil’s public health system based on financial perspectives, clients, internal processes, learning, and growth. In the second, the authors presented a methodology to measure the successes and failures of health systems, encompassing economic, educational, cultural, geographic, and epidemiological components. The last two studies analyzed the response capacities of the health systems in Hong Kong and Germany, respectively, using simple, multiple, and logistic regression.

In addition, several studies have assessed the efficiency of health systems or their components on the basis of the Data Envelopment Analysis (DEA) model. For example, Botega et al. [10] used DEA to evaluate the economic efficiency of Brazilian general hospitals. Jing et al. [11] assessed the technical efficiency of public and private hospitals in Beijing, and Alatawi et al. [12] evaluated the performance of public hospitals in Saudi Arabia, detecting the sources of inefficiency.

However, despite the efforts identified in the aforementioned studies, the complex problems of healthcare, such as assessing the performance of its systems, are multidimensional and require multi-criteria methods to solve them. Mühlbacher and Kaczynski [13], for example, conducted a review in order to identify current health research and showed that decision problems in this area are characteristically of the multi-criteria variety. Studies have focused on developing techniques to weigh and score different decision criteria.

Indeed, in decision-making processes for complex, poorly-structured problems with several potentially conflicting assessment criteria and individual preferences, it is natural to resort to decision-making techniques with multiple criteria [13,14,15,16,17].

In Brazil, one of the tools used by the Brazilian government to assess the Unified Health System (SUS) is the Index of Unified Health System Performance (IDSUS), which measures access to health services and the effectiveness of these systems; that is, it functions as a representation of the country’s health system [18]. The IDSUS considers 24 performance indicators, of which 14 are related to healthcare access and 10 are related to the effectiveness of the service [19]. It is calculated by the weighted sum of the indicators analyzed and principal component analysis (PCA) to measure the weights of the indicators [20].

However, it is important to underscore the limitations of the IDSUS since it assesses the performance of a health system, which is a multi-criteria problem, as previously stated. It is an index calculated by the weighted sum of indicators that excludes decision makers in the weight-measuring process. This is concerning because the weight distribution process is an essential stage in multi-criteria models, and the ideas and priorities of decision makers should be considered [21] since they have a direct influence on decision making [22,23].

Thus, given the importance of the Unified Health System (SUS), the aim of the present study was to propose a model based on the R-Technique for Order of Preference by Similarity to Ideal Solution (R-TOPSIS) in order to assess the performance of the SUS. In summary, the main aims of the present study were (i) to propose a multi-criteria model to assess the performance of the SUS in close collaboration with healthcare decision makers; (ii) to validate of the proposed model with a real case study, highlighting its ability to replicate and the dynamicity required for the healthcare context; and (iii) to compare of the results obtained with other assessment proposals.

## 2. Unified Health System (SUS)

Healthcare in Brazil is divided into public and private systems, and all Brazilians enjoy free access to the services of the Unified Health System (SUS), which operates through federal, state, and municipal institutions, in addition to private entities that provide services to the SUS by means of agreements with public bodies [24,25].

The construction of the SUS in the last three decades has been unique in Latin America [26] and is currently considered a model for the rest of the world [27,28]. Its origin lies in the Brazilian Federal Constitution of 1988, which stipulates that health is the responsibility of the government and that every citizen has the right to free and comprehensive healthcare [27,28,29]. In this system, services are decentralized and hierarchically organized, with municipalities performing a central role, according to the principles of universality, all-encompassing care, and equal access [10,30].

According to Aly et al. [31], the Ministry of Health conducts systematic assessments and coordinates health and performance indicators of the Brazilian healthcare system by contributing to strategic and operational analyses and monitoring and assessing the fulfillment of SUS objectives. To that end, the Performance Index of the SUS (IDSUS) was created in 2012 as a participatory process and important management support initiative.

The IDSUS assesses healthcare on the basis of 24 simple and compound indicators capable of measuring potential or actual access and the effectiveness of actions and health services at the different care levels (basic, specialized outpatient and hospital, and urgencies and emergencies). Its construction follows the structure outlined in its founding document, available from the Ministry of Health [20].

According to the available assessment model, the IDSUS was formed using statistical methods, parameters, and scores. With respect to the first, in order to calculate simple indicators, the following methodologies were applied: indirect standardization by age group and sex, with the goal of decreasing the influence of the different quantitative compositions of age range and sex that exist between the populations of the municipalities; empirical Bayes, which seeks to reduce the effect of the variation in the results of the indicators in small populations; and the 3-year average, which is the only additional way of overcoming the small numbers and calculating all the indicators for municipalities with small populations.

For compound indicators or access indices and effectiveness by the level of healthcare, the assessment used the simple linear correlation test, in which the simple indicators that exhibited the highest correlation (positive or negative) were selected. To attribute weights to the simple indicators, aiming to achieve the access indices and effectiveness by the level of healthcare, principal components analysis (PCA) was adopted. This is a multivariate statistical technique used to transform original variables into others with the same dimension, losing as little information as possible.

The IDSUS used nationally and internationally recognizes parameters as a type of reference for the desired goals. Finally, the IDSUS score is a proportion of the result in relation to the parameter.

In summary, in order to assess the SUS, the 24 indicators were calculated, and the results were divided by their respective parameters, obtaining the score for each indicator. Weights were attributed to these scores using the PCA method, resulting in the compound indicators or potential or obtained access indices and effectiveness indices at the different healthcare levels. Weights were also assigned to the indices by the level of healthcare to produce the respective access and effectiveness indices of the SUS, which, in turn, resulted in the performance index of the SUS, namely, the IDSUS.

## 3. Materials and Methods

In line with the procedure to construct MCDA models [32], this study followed three main phases (Figure 1); the first two are presented in this section, and the third is presented in Section 4.

In the preliminary phase, with the aim of analyzing the performance of the SUS and the opinions of two decision-makers linked to Brazil’s public health system, the assessment criteria were defined. These criteria were based on the elements of the IDSUS [20]. The sub-criteria, criteria, and categories are presented in Table 1.

Each sub-criterion was scored according to the proportion of the result in relation to its respective parameter, as shown in Table 1 and following the same logic as the IDSUS [20]. The result of the indicator, therefore, was the result of the city divided by the parameter of the indicator. This quotient, multiplied by 10, resulted in a score from 0 to 10 for each simple sub-criterion and showed the difference between the actual and desired situation.

Although the model is universal, a set of six cities (alternatives) was established to better exemplify model usage, comprising Campinas ($A_{1}$), Ribeirão Preto ($A_{2}$), Santos ($A_{3}$), São José do Rio Preto ($A_{4}$), São Paulo ($A_{5}$), and Sorocaba ($A_{6}$). These were chosen because of the relevance of the cities and the state of São Paulo—the most socially and economically developed in Brazil. Their Human Development Index (HDI) is considered high (0.783), their per capita Gross Domestic Product (GDP) is the highest in the country, and the GDP corresponds to 32% of the country’s total [33].

The homogeneity of the cities was also considered. Homogeneity is a resource used by the IDSUS to group cities according to socioeconomic similarities, child mortality profile, and the health system structure existing in the cities [20].

On the basis of these preliminary definitions, the structure (P; I) was defined in the second phase of preference modeling and method choice, provided that a complete ordering of the alternatives could be obtained in this structure. This is the traditionally preferred model, with which many MCDA methods are associated. Given that the compensation between the criteria is permitted in the problem analyzed, compensatory rationality was indicated. For this case, an extension of the TOPSIS was used (R-TOPSIS).

The TOPSIS method is characterized by its easy use and robust results, which have led to its widespread application, as reported by Behzadian et al. [34]. Nevertheless, the TOPSIS has been criticized for the problem of Rank Reversal (RR). RR refers to the change in the rank ordering of some alternatives after an alternative has been added or excluded from this previously ranked group [35]. This phenomenon has been debated for over 30 years and for different MCDM methods.

In order to resolve this problem for the TOPSIS, Aires and Ferreira [36] proposed the R-TOPSIS. As their primary premise, the authors considered that changes in the original method should be minimal to make the new method easier for users of the TOPSIS method and maintain compatibility and rationality between them. Thus, the authors proposed two changes to the original TOPSIS method, as follows:The use of an additional input parameter called a domain, i.e., a numerical value (integer or real) that represents the range of possible values that each criterion could take;A change in the normalization procedure. The R-TOPSIS uses Max–Min normalization or Max normalization to fix the ideal solutions and ensure that there is no change in the values of the normalized and weighted decision matrices after modifications are introduced to the initial decision problem.

On the basis of the changes proposed, the method proved to be robust and immune to the different RR cases presented in the literature when submitted to numerous simulated decision problems and a real student selection case; see Aires et al. [37].

This method is especially relevant for the analysis of the problem discussed in this paper since it can be characterized as a decision-making problem in a dynamic context [38], in which new cities can change the assessment. In this context, RR problems are extremely undesirable.

The swing weight procedure was used to establish the weights of the categories, criteria, and sub-criteria [39]. In order to more realistically model decision-making problems, elicitations are based on changing attributes or the direct attribution of weight intervals [40]. This is because decision makers can easily attribute weights and overcome the problem identified in the IDSUS, as previously mentioned. The swing method is widely used and accepted in the healthcare field, as reported by Beynon and Kitchener [41], Medeiros and Ferreira [42], and Németh et al. [43].

Two healthcare specialists were consulted regarding weight attribution. In this procedure, first, a hypothetical situation is defined as the worst possible hypothesis for all the categories, criteria, and sub-criteria [44,45]. Thus, a value of 0 was established for all the cases, and the process was performed initially for the sub-criteria analyzed, followed by the criteria and finally the categories.

The specialists were then consulted about which of the sub-criteria in each of the criteria was the most important, given the performance of the SUS. The best assessed received a score of 100, and the others were defined proportionally according to their opinions. The final weight of each sub-criterion was calculated on the basis of the final weight of each sub-criterion, calculated by dividing its score by the sum of the scores of all the sub-criteria. Each decision maker made an assessment, and the final weights were the averages of the individual evaluations. The results of the sub-criteria, criteria, and categories are presented in Table 2.

Finally, Section 4 presents the finalization phase, in which the alternatives were assessed by applying the decision model. In this phase, a sensitivity analysis was also conducted, and recommendations were made.

## 4. Results and Discussion

As mentioned earlier, the present study selected the following six cities (alternatives) to better exemplify the use of the model: Campinas ($A_{1}$), Ribeirão Preto ($A_{2}$), Santos ($A_{3}$), São José do Rio Preto ($A_{4}$), São Paulo ($A_{5}$), and Sorocaba ($A_{6}$). The decision matrix data were collected from IDSUS reports [20] and are presented in Table 3.

The R-TOPSIS was applied on the basis of the data in Table 3. After the decision matrix was defined, the values 10.0 and 0.0 were stipulated as the domains of all the criteria, given that these were the extreme values for all of them. Next, normalization was performed using the Max. The results are presented in Table 4 in terms of the distances of each alternative for the ideal positive (DPIS) and negative (DNIS) solutions, closeness coefficient (CC), and ranking order.

The results in Table 4 show that the city of Ribeirão Preto ($A_{2}$) exhibited the best performance, and Santos ($A_{3}$) had the worst performance. These results are explained by the fact that the city of Ribeirão Preto is the alternative that presented the best performance in the category of effectiveness, and its criteria and sub-criteria added up to 0.600 of the weight of the analyzed problem. This category is of great importance in the model since it is responsible for measuring the quality of the services provided, understood as the characteristic that shows the degree to which services and actions achieve the expected results [20].

Likewise, the results obtained for the city of Santos are explained by its low performance in the same category (second worst among the alternatives analyzed), along with its low performance in the category of potential or obtained access (worst among the alternatives analyzed). Potential access makes it possible to infer the potential supply of care, while the access obtained is measured through the actual care provided; that is, it is a category that corresponds to the capacity of the health system to guarantee the necessary care promptly and with adequate resources [20]. This highlights the low performance in the Medium-Complexity Outpatient and Hospital Care criterion and, mainly, the High-Complexity Outpatient and Hospital Care Reference for Medium- and High-Complexity and Urgency and Emergency Care criterion, whose results were much lower than those of the others.

Comparing the results obtained with those of other studies in the area is a challenge since the analysis proposed here was not considered in other studies. This is reinforced by the findings of Adunlin et al. [3] since none of the 66 articles analyzed in their literature review presented the same proposal. Despite this, it is possible to discuss some studies that have similar aspects to this one.

Gramani [6] also investigated the performance of Brazil’s public health system, but they focused on different perspectives (financial, clients, internal processes, learning, and growth) and used a different approach (DEA). Likewise, Botega et al. [10] used the DEA to assess the economic efficiency of Brazilian general hospitals that provide inpatient care to the Unified Health System (SUS). DEA was also used by Jing et al. [11] to assess the technical efficiency of public and private hospitals, but in Beijing, China.

Other studies were also carried out in different countries using regression techniques. This was the case for Gerring et al. [7], who presented a methodology to measure the successes and failures of health systems, encompassing economic, educational, cultural, geographic, and epidemiological components. Vandan et al. [8] compared the health system responsiveness perceived by ethnic minority people in South Asia with that perceived by local Chinese people in Hong Kong. Finally, Tille et al. [9] identified the general levels of responsiveness of the health system and the determinant associations of outpatient healthcare in Germany from the user’s perspective.

Thus, the original and innovative contribution of the proposed model is to serve as a basis for other studies in the Brazilian context and in different countries to carry out similar analyses using MCDA/M methods. This is in line with what Mühlbacher and Kaczynski [13] highlighted, as there is a need for more research to develop practical guidelines for the proper application and reporting of MCDA/M methods.

In addition, different from the observations here, the IDSUS results found that São José do Rio Preto ($A_{4}$) was the best city in São Paulo state. This city ranks only fifth (next to last) in the proposed model, demonstrating how decision makers’ opinions affect the results.

This, as already pointed out, is concerning because the weight distribution process is an essential stage in multi-criteria models since they have a direct influence on decision making [22,23]. Therefore, it is clear that special attention should be given to defining the criteria weights [46], as highlighted by Vavrek [47], who focused on clarifying the impact of determining the criteria weights, using TOPSIS as a basis.

Furthermore, as one of the aims of the present study was to propose a multi-criteria model to assess the performance of the SUS in close collaboration with healthcare decision makers, special attention was paid to the procedure for determining the weights of the model. In this case, the swing weight procedure proved to be effective, as the specialists in the study found the method relatively easy. This factor was pointed out by Zardari et al. [48] as one of the advantages of this type of procedure. Several other advantages were also reported: (i) it is an efficient and effective way to discuss, evaluate, summarize, and explain attribute weights; (ii) it is relatively simple and transparent; (iii) it is more parsimonious than techniques involving pairwise comparisons when many (>4) criteria need to be weighted; and (iv) it is quite fast, and respondents respond promptly [48].

Finally, the use of an appropriate approach for the problem, as is the case for MCDA/M, had an impact on the results since when the proposed approach was applied, but with weights originally used in the IDSUS, São José do Rio Preto ($A_{4}$) ranked only fourth. Table 5 shows a comparison between the results.

The results of the proposed model are quite different from those of the IDSUS. This shows the importance of using appropriate approaches to the analyzed problem. After all, great business and government decision makers are interested in solutions that obtain the best results for their companies and their countries. It is not by chance that multi-criteria decision analysis is at the heart of many crucial problems today [16]. Its origin is related to the success of operations research, which emerged during the Second World War and, due to its great success, began to be incorporated by companies. Then, in the 1970s, the first methods aimed at problems in the multi-criteria environment appeared.

These methods brought to light the importance of the human factor, something that was crucial for the model proposed in this study. MCDA/M methods provide greater understanding, on the part of the actors involved in the decision-making process, of the different dimensions of the problem.

This becomes clear when it is noted that multi-criteria decision analysis is becoming an increasingly popular framework to assist and support decision making in healthcare. Adunlin et al. [3] corroborated this by pointing out that the MCDA/M has the potential to improve decision making in this area and highlighting that it has been considered by several public and private health organizations and agencies, including the US Agency for Healthcare Research and Quality’s (AHRQ), the UK Department of Health, the National Institute for Health and Care Excellence (NICE) in England and Wales, the UK Office for Health Economics (OHE), among others.

Therefore, although decision making in healthcare does not differ conceptually from decision making in other fields, there is an important difference in this area from the others—the fact that health is an irreplaceable and priceless good, as highlighted by Diaby et al. [49]. The authors also added that it is more difficult for healthcare decision makers to make the right choices, as their decisions have enormous consequences on the quality of life and society as a whole.

In summary, the main contribution of this study is that it provides a model capable of helping decision making in the healthcare area. Considering the Brazilian context and the importance of its public health system to society, assessing the performance of the SUS with adequate parameters enables resources to be distributed on the basis of robust and reliable results.

According to Silva et al. [28], almost 75% of the population only uses the SUS when they need healthcare. However, public funding was never sufficient to achieve the goal of a universal system that would reduce social inequalities [26].

This was reflected in the results of the study, as within the same state, two cities presented very discrepant results in terms of effectiveness related to the quality of services provided, as well as in terms of access to services related to the capacity of the system of health in ensuring the necessary care promptly and with adequate resources.

Thus, as obstacles to structural change persist with sustained effects on health inequalities [26], the proposed model appears to be an additional decision-making support tool for a healthcare system that faces complex challenges.

Finally, according to Almeida et al. [32], a number of recommendations can be made regarding the usefulness of the proposed model. The results are simplifications of the problem modeling process, such that qualitative analyses of each parameter, as well as the other quantitative models, are also important. The solution obtained was reliable, but the nature of the model proposed here should be taken into account. The aim of the study was not to suggest that this is the only correct model but rather to demonstrate its usefulness.

### Sensitivity Analysis

Sensitivity analysis was conducted to assess the impact caused by the 10 and 20% higher or lower variation in the weights of the stability sub-criteria of the final classification. As one of the weights increased or decreased, its difference was equally distributed over the rest of the sub-criteria.

With respect to the 10% variation, no impact was observed; that is, the order did not change in any situation. However, for 20%, when the weights decreased and increased, stability was, on average, 96.30 and 94.44%, respectively. The only classification changes occurred when the sub-criteria $C_{15}$ and $C_{16}$ increased and decreased, as shown in Figure 2.

Figure 2 shows the following changes in rankings with an increase and decrease in weights:Alternatives $A_{1}$ and $A_{6}$ inverted their rankings when the weight of sub-criterion $C_{15}$ changed;Alternatives $A_{5}$ and $A_{6}$ inverted their rankings when the weight of sub-criterion $C_{15}$ decreased;Alternatives $A_{1}$ and $A_{6}$ and $A_{4}$ and $A_{3}$ inverted their rankings when the weight of sub-criterion $C_{16}$ increased;Alternatives $A_{4}$ and $A_{5}$ inverted their rankings when the weight of sub-criterion $C_{16}$ changed.

This is due to the fact that these were the two sub-criteria with the highest weight in the assessment. As such, when the variation was greater for these sub-criteria, the results were more susceptible to change. Finally, it is important to underscore that the highest ranking did not change in any of the scenarios analyzed, confirming good model stability. Thus, the rankings obtained showed an optimal stability of 100.0 and 95.37% in response to 10 and 20% changes in the weights of sub-criteria, respectively.

## 5. Conclusions

Brazil’s Unified Health System was assessed using a multi-criteria approach based on TOPSIS. The proposal of integrating multiple criteria, as well as considering healthcare decision makers, provides a clearer picture of the object investigated. The results also demonstrate that the R-TOPSIS method is more reliable for including new cities in the analysis without the risk of undesirable inversions, in addition to allowing replications.

This study contributes to the research on the performance of health systems, presenting results on the general assessment of the SUS through a multi-criteria model for the first time. Our findings can serve as a resource for health decision makers in Brazil regarding the distribution of resources to cities and states with lower rates of access and effectiveness of health services. More precisely, they can assist in decision making about priorities. Furthermore, the approach is applicable to any set of alternatives that needs to be evaluated, which is important for decision makers involved in health management.

In summary, the main findings of the study were (i) the proposal of a multi-criterion model to assess the performance of the SUS, in close collaboration with healthcare decision makers; (ii) the validation of the proposed model with a real case study, highlighting its ability to replicate and the dynamicity required for the healthcare context; and (iii) a comparison of the results obtained with other assessment proposals.

The study has some limitations, some of which are considered opportunities for future studies. The first concerns the sample, since only six cities in the state of São Paulo were considered for model validation. This is because the homogeneity of the cities and the ease of presenting the results were taken into account. However, this is a limitation that can be easily addressed in future studies, given the dynamic nature of the model.

Second, the use of the MCDA/M for the development of the model may raise doubts in those interested in this study, especially those who are not familiar with these methods. This was also highlighted in the literature review by Mühlbacher and Kaczynski [13].

Third, the proposed model included aspects directly linked to the access and effectiveness of SUS health services, an innovative and unique contribution. However, other dimensions were not analyzed, such as user satisfaction and financial issues, as carried out by Botega et al. [10], or geographic inequalities, pointed out by Castro et al. [27] as being of great relevance to the Brazilian context.

Finally, in addition to the suggestions for future studies presented based on the aforementioned limitations, it is suggested that further research be carried out to develop practical guidelines for the appropriate application and reporting of MCDA/M methods in the evaluation of health system performance. It is also suggested to use the proposed model in different countries and with different weighting procedures.

## Figures and Tables

**Figure 1 ijerph-19-11478-f001:**
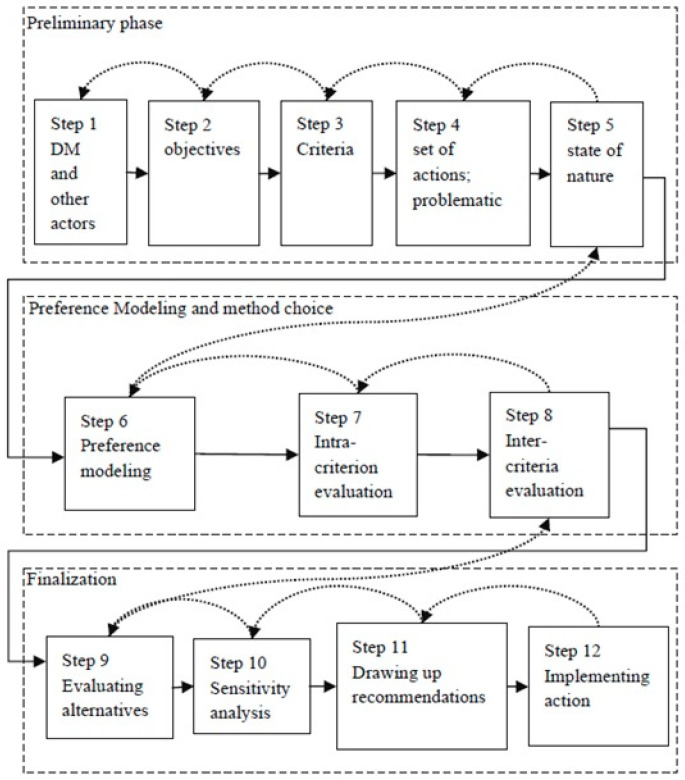
Procedure for resolving an MCDM/A problem.

**Figure 2 ijerph-19-11478-f002:**
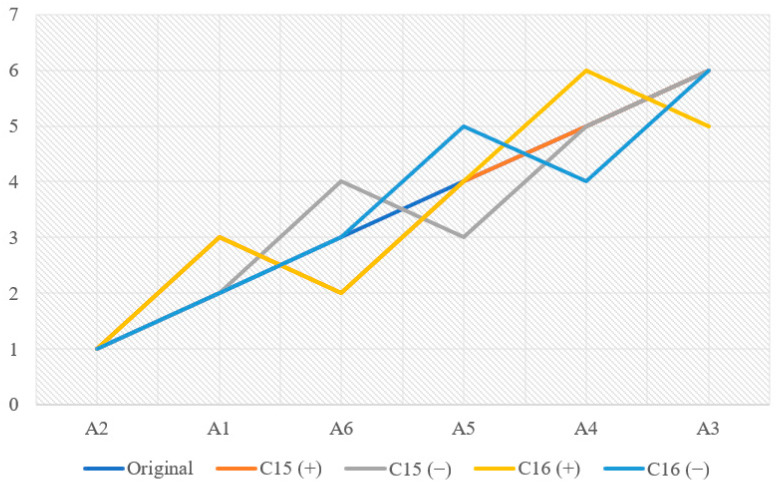
Sensitivity analysis.

**Table 1 ijerph-19-11478-t001:** Criteria and sub-criteria.

Category	Criterion	Sub-Criterion	Abbreviation	Estimation Parameter
Potential or Obtained Access	Basic Care	Population coverage estimated by basic health teams	C_1_	100%
		Population coverage estimated by basic oral health teams	C_2_	50%
		Proportion born live from mothers with 7 or more prenatal consultations	C_3_	90%
	Medium-Complexity Outpatient and Hospital Care	Ratio of mammograms performed on women aged 50 to 69 years and the population in the same age range	C_4_	70 examinations for every 100 women in two years
		Ratio of cytopathological examination of the uterine cervix in women aged 25 to 59 years and the population in the same age range	C_5_	90 examinations for every 100 women in 3 years
		Ratio of medium-complexity clinical–surgical hospitalizations and the resident population	C_6_	6.3 hospitalizations/100 inhabitants per year
		Ratio of medium-complexity outpatient procedures selected and the resident population	C_7_	2.6 procedures/100 inhabitants per year
	High-Complexity Outpatient and Hospital Care Reference for Medium- and High-Complexity and Urgency and Emergency Care	Ratio of high-complexity clinical–surgical hospitalizations and the resident population	C_8_	6.3 hospitalizations/100 inhabitants per year
		Ratio of high-complexity outpatient procedures selected and the resident population	C_9_	7.8 procedures/100 inhabitants per year
		Proportion of hospital access of accidental deaths	C_10_	70%
		Proportion of medium-complexity hospitalizations performed for non-residents	C_11_	0.72%
		Proportion of high-complexity hospitalizations performed for non-residents	C_12_	1.14%
		Proportion of high-complexity outpatient procedures performed for non-residents	C_13_	1.17%
		Proportion of medium-complexity outpatient procedures for non-residents	C_14_	0.90%
Effectiveness	Basic Care	Basic Care Effectiveness Index	C_15_	“Proportion of basic care hospitalizations” minus 0.15 for each indicator point lost in “Incidence rate of congenital syphilis” and “Proportion of new bacilipherous pulmonary tuberculosis cases cured” minus 0.1 for each indicator point lost in “Proportion of new Hansen’s disease cases cured” and “Vaccine coverage with the tetravalent vaccine”
	Medium- and High-complexity, Urgency, and Emergency	Proportion of normal deliveries	C_16_	70%
		Proportion of deaths in the intensive care unit of individuals younger than 15 years old	C_17_	10%
		Proportion of deaths in those hospitalized for acute myocardial infection	C_18_	10%

**Table 2 ijerph-19-11478-t002:** Criteria weight.

Specialist 1			Specialist 2			Final Weight		
Categories	Criteria	Sub-Criteria	Categories	Criteria	Sub-Criteria	Categories	Criteria	Sub-Criteria
0.355	0.169	0.072	0.444	0.185	0.082	0.400	0.177	0.077
		0.032			0.045			0.039
		0.065			0.058			0.061
	0.084	0.029		0.148	0.049		0.116	0.039
		0.026			0.042			0.034
		0.006			0.017			0.012
		0.024			0.040			0.032
	0.101	0.027		0.111	0.030		0.106	0.028
		0.024			0.027			0.025
		0.008			0.009			0.008
		0.016			0.018			0.017
		0.007			0.007			0.007
		0.005			0.006			0.006
		0.014			0.015			0.014
0.645	0.430	0.430	0.556	0.208	0.208	0.600	0.319	0.319
	0.215	0.092		0.347	0.148		0.281	0.120
		0.041			0.081			0.061
		0.082			0.118			0.100

**Table 3 ijerph-19-11478-t003:** Decision matrix.

Alt	C_1_	C_2_	C_3_	C_4_	C_5_	C_6_	C_7_	C_8_	C_9_	C_10_	C_11_	C_12_	C_13_	C_14_	C_15_	C_16_	C_17_	C_18_
A_1_	5.45	5.18	9.18	3.62	6.19	4.31	5.05	5.07	8.00	9.07	10.00	10.00	10.0	10.00	9.19	5.29	10.00	6.75
A_2_	4.13	5.81	8.72	3.80	5.37	5.89	3.82	7.77	8.91	7.92	10.00	10.00	10.0	7.86	9.85	5.91	10.00	10.00
A_3_	3.48	5.87	9.42	3.08	6.79	3.64	3.15	2.98	5.99	8.93	7.23	3.20	10.0	1.72	9.03	4.86	8.89	5.75
A_4_	4.31	5.99	9.80	5.37	5.96	7.94	7.25	10.00	10.00	7.21	10.00	9.75	10.0	7.32	9.84	2.23	9.68	7.55
A_5_	4.60	3.42	8.46	3.68	5.84	4.81	5.28	5.31	7.79	10.00	10.00	10.00	10.0	10.00	8.85	6.74	10.00	6.03
A_6_	3.43	4.15	10.00	5.13	6.46	4.24	3.50	4.17	6.04	9.51	6.49	5.31	10.0	4.79	9.91	6.31	5.95	5.46

**Table 4 ijerph-19-11478-t004:** Results.

Alt	DPIS	DNIS	Closeness Coefficient	Ranking
A_1_	0.0883	0.3254	0.7865	2
A_2_	0.0781	0.3525	0.8187	1
A_3_	0.1071	0.3141	0.7457	6
A_4_	0.1098	0.3424	0.7572	5
A_5_	0.0906	0.3163	0.7774	4
A_6_	0.0958	0.3414	0.7808	3

**Table 5 ijerph-19-11478-t005:** Comparison of the results.

Alt	Proposed Model		Proposed Model with IDSUS Weights		IDSUS	
	CC_i_	Ranking	CC_i_	Ranking	Index	Ranking
A_1_	0.7865	2	0.5389	3	6.0585	3
A_2_	0.8187	1	0.5564	1	6.3170	2
A_3_	0.7457	6	0.4715	5	5.1631	6
A_4_	0.7572	5	0.5288	4	6.4708	1
A_5_	0.7774	4	0.5471	2	6.0281	4
A_6_	0.7808	3	0.5320	4	5.5698	5

## Data Availability

Not applicable.

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
