# Peer review of "A Multi-Criteria Approach to Assess the Performance of the Brazilian Unified Health System"

_ijerph, 2022, doi:10.3390/ijerph191811478_

Round 1
Reviewer 1 Report
Dear authors,
Thank you for allowing me the opportunity to review the manuscript titled “A multi-criteria approach to assess the performance of the Brazilian Unified Health System”.
From my point of view, the authors study asses the performance of the Brazil´s Unified Health System in a set of six Brazilian cities (not randomly selected), through the TOPSIS technique, comparing it with the Index of Unified Health System Performance.
The main study strength is in my opinion, proposing an alternative to assess a country health system with a real case in the practice.
Nevertheless, the study has some flaws and issues that need to be adressed before considering for publication. Here you have my feedback,
Major issues,
Abstract could be rewritten in an structured and weighted form, following the classical sections, introduction (shortly, with the main aim), methodology, main results and conclusions. Also in the abstract, I wonder if the “method proposed (line 20) is TOPSIS?
Keywords, Please, make sure that the words /achronims SUS, IDSUS are universally and useful to do a bibliography search.
Please, define the study design in the methodology section. It is mentioned “validation of the proposed model with a real case study” but I suggest to concrete the kind of study done by authors.
Also in methodology it is not clear if authors are comparing two techniques or methods (IDSUS vs TOPSIS or MDCA?). It seems to me that the reader catch this idea reading the beginning of the abstract.
The “Discussion” section is missed. I suggest authors to review the “Instructions for authors” document, in order to draw up a discussion section, or combined it with results section, where authors discuss the results and how they can be interpreted of previous studies and of the working hypotheses.
Perhaps some paragraphs from the other sections could be “move” to the discussion one.
Also, I miss a paragraph about the study limitations, beyod the “limitations of IDSUS” mentioned by authors. For example, the selection bias, unless the six Brazilian cities where randomly selected.
I also suggest to review the conclusions section in order to be re-written.
Minor issues
In abstract authors mention that “results were also compared with differen approaches…” But, with which ones?
Please, define the TOPSIS acronym (Technique for Order of Preference by Similarity to ideal Solution) the first time that it is mentioned during the manuscript, in order to facilitate the text reading to people who ignore this term.
Introduction
Its seems to me too long. Perhaps some content could move to other study sections.
In line 59 authors mention that IDSUS is “one of the tool used by the Brazilian government to assess the Unified Health System”. But, which ones are the other tools? Is it IDSUS the main or the “oficial” tool?
Lines 75-79. Instead of “main contributions” it looks like they are the study aims.
Lines 80-84 paragraph is, in my opinion, superfluous.
As I have revised in the literature that IDSUS assess wether the population is getting to be answered in the public network and quality of service in all municipalities, states and at the Brazil national level according tos cale 0 to 10. You have it in DOI: 10.14422/rib.i05.y2017.006
In case you are comparing a MDCA model like R-TOPSIS (is this correct?) versus IDSUS, perhaps some paraghps about IDSUS would be useful to discuss the alternative method that authors proposed.
Lines 126-132 seems to me a bit confusing. I understand that authors are using the PCA method resulting in IDUS. In any case, it would belong to the methodology section.
Materials and Method
Table 1. Please, define ICU (Intensive Care Unit)
Line 153, Please, define HDI and GDP, and RR in line 183.
Lines 183-200 I understand that the reference 37 describe the steps of the R-TOPSIS. Then I suggest to eliminate those lines, or considering it like supplementary material (in case it is original).
Lines 201-211 could be in discussion, in the broadest context posible.
Line 223 and Table could be move to the “results” section.
Results
Lines 256-269 could be in the “Discusion” section, except the sentence “the main contribution of the study is to provide a model capable of helping decision making in the healthcare area” which I see as a conclusion; and the lines 267-269 where authors express “The aim of the study…(to demonstrate its usefulness).
Lines 270-273 I see more convenient this first paragraph in “Material and methods” section, and the rest in “Results”.
Conclusions
I suggest to re write it. For example, the first sentence in line 298 is not a conclusion from my point of view. Paragraph in lines 308-312 looks like aims.
Why “R-TOPSIS method is more reliable for including new cities”? (line 301)
Finally, the last sentence could be at the end of the “Discussion” section (“may be useful”), and as future research directions.
Author Response
Reviewer #1 Comments
Reviewer:
- Abstract could be rewritten in an structured and weighted form, following the classical sections, introduction (shortly, with the main aim), methodology, main results and conclusions. Also in the abstract, I wonder if the “method proposed (line 20) is TOPSIS?
Response: Thank you for the observation. We have restructured the abstract following the recommendations and correcting the problems.
- Keywords, Please, make sure that the words /acronyms SUS, IDSUS are universally and useful to do a bibliography search.
Response: Thank you for the observation. The keywords used follow the main studies in the area, which mention acronyms in the same way. See, for example, the following studies:
Castro, M.C.; Massuda, A.; Almeida, G.; Menezes-Filho, N.A.; Andrade, M.V.; Noronha, K.V.M.S; Rocha, R.; Macinko, J.; Hone, T.; Tasca, R.; Giovanella, L.; Malik, A.M.; Werneck, H.; Fachini, L.A.; Atun, R. Brazil’s unified health system: the first 30 years and prospects for the future. The Lancet 2019, 394, 345–356.
Paim, J.; Travassos, C.; Almeida, C.; Bahia, L.; Macinko, J. The Brazilian health system: history, advances, and challenges. The Lancet 2011, 377, 1778–1797.
Dantas, M.K.; Oliveira, L.R.; Ferolla, L.M.; Paschoalotto, M.A.C.; Lopes, J.E.F.; Passador, J.L.; Passador, C.S. Cross-sectoral assessment of public policies in health and the environment: scenario of the municipalities in the state of Sao Paulo. Eval Program Plann. 2017, 65, 30-39.
Araújo, C.; Barros, C.P.; Wanke, P. Efficiency determinants and capacity issues in Brazilian for-profit hospitals. Health Care Manag Sci 2014, 17, 126–138.
Machado, C.V.; Silva G.A. Political struggles for a universal health system in Brazil: successes and limits in the reduction of inequalities. Globalization and Health 2019, 15, 77.
Gramani, M.C. Inter-Regional Performance of the Public Health System in a High-Inequality Country. PLoS ONE 2014, 9, e86687,
Botega, L.A.; Andrade, M.V.; Guedes, G.R. Brazilian hospitals' performance: an assessment of the unified health system (SUS). Health Care Manag Sci. 2020, 23, 443-452.
- Please, define the study design in the methodology section. It is mentioned “validation of the proposed model with a real case study” but I suggest to concrete the kind of study done by authors. Also in methodology it is not clear if authors are comparing two techniques or methods (IDSUS vs TOPSIS or MDCA?). It seems to me that the reader catch this idea reading the beginning of the abstract.
Response: Thank you for the observation. We would like to clarify that in articles involving MCDM, it is common to use the term "validation" when a case study is carried out to demonstrate the operability/reliability of the model. Then, a case study was carried out that helped to validate the proposed model, as highlighted by the reviewer. On the second question, the primary aim of the study was to develop a model, which was designed given the limitations of IDSUS. Thus, to show the contribution of the proposed model (based on a multi-criteria logic), we also compared it with IDSUS, as a secondary aim. Finally, as we rewrote the abstract, as per the answer to item 1, we hope this is now clear.
- The “Discussion” section is missed. I suggest authors to review the “Instructions for authors” document, in order to draw up a discussion section, or combined it with results section, where authors discuss the results and how they can be interpreted of previous studies and of the working hypotheses. Perhaps some paragraphs from the other sections could be “move” to the discussion one.
Response: Thank you for pointing out this problem. We have restructured the results and added the discussion in this section. We believe the section has become much more robust.
- Also, I miss a paragraph about the study limitations, beyod the “limitations of IDSUS” mentioned by authors. For example, the selection bias, unless the six Brazilian cities where randomly selected. I also suggest to review the conclusions section in order to be re-written.
Response: Thank you for pointing out this problem. We rewrote the conclusions section, including presenting several limitations of our study.
- In abstract authors mention that “results were also compared with different approaches…” But, with which ones?
Response: Thank you for the observation. As shown in Table 5 and later discussed, the result of the proposed model was compared with two other approaches: the IDSUS (original) and the proposed model, but with the weights originally used by IDSUS. This comparison was made to emphasize how the use of an appropriate approach to the problem, as is the case of MCDA/M, and the opinions of decision-makers, have an impact on the results.
- Please, define the TOPSIS acronym (Technique for Order of Preference by Similarity to Ideal Solution) the first time that it is mentioned during the manuscript, in order to facilitate the text reading to people who ignore this term.
Response: Thank you for this reminder. We made the definition.
- Introduction: Its seems to me too long. Perhaps some content could move to other study sections.
Response: Thanks for the suggestion. As we have carried out a thorough review of the results and conclusions, in addition to having included the discussion in the study, if the reviewer agrees, we would like to keep the introduction as it is currently presented, as we believe that it follows a well-founded and clear construction logic
- Introduction: In line 59 authors mention that IDSUS is “one of the tool used by the Brazilian government to assess the Unified Health System”. But, which ones are the other tools? Is it IDSUS the main or the “oficial” tool?
Response: Thank you for the observation. Given the complexity of the Brazilian health system, the government uses some tools while seeking to develop new ones. Tools such as the National Program for Evaluation of Health Services (PNASS), whose main aim consists of evaluating the satisfaction of users concerning emergency services, hospitalization, and outpatient clinic, and the Project for the Development of a Methodology for the Evaluation of the Performance of the Brazilian Health System (PROADESS), which aims to develop and implement a methodology to evaluate the performance of health services at the national level, are other examples.
However, IDSUS, an official assessment tool created by the Ministry of Health of Brazil, constitutes a more complete instrument for dealing with the investigation of the main deficiencies in the health framework in the municipalities of Brazil, indicating the points critical, whose results can support decision-making. Therefore, the article presented an approach that extends IDSUS, solving its limitations.
- Introduction: Lines 75-79. Instead of “main contributions” it looks like they are the study aims. Introduction: Lines 80-84 paragraph is, in my opinion, superfluous.
Response: Thank you for the suggestion. We changed the “main contributions” to “main aims” and removed the lines 80-84 paragraph.
- Introduction: As I have revised in the literature that IDSUS assess wether the population is getting to be answered in the public network and quality of service in all municipalities, states and at the Brazil national level according tos cale 0 to 10. You have it in DOI: 10.14422/rib.i05.y2017.006
Response: Thank you for the observation. This article reinforces the importance of IDSUS, as pointed out in the previous answer, as well as pointed out in section 2 of our article.
- Introduction: In case you are comparing a MDCA model like R-TOPSIS (is this correct?) versus IDSUS, perhaps some paragraps about IDSUS would be useful to discuss the alternative method that authors proposed.
Response: Thank you for the suggestion. The article aims to present an MCDM approach that extends IDSUS by overcoming its limitations. So, there is a comparison between the results obtained in both. The main limitations of IDSUS are presented in lines 67-73 of the introduction. So that the introduction doesn't get too long, details about IDSUS were quoted in section 2, lines 99-129. We believe that this makes reading easier, but we make ourselves available to the reviewer to make any changes.
- Introduction: Lines 126-132 seems to me a bit confusing. I understand that authors are using the PCA method resulting in IDSUS. In any case, it would belong to the methodology section.
Response: Thank you for the observation. No, we are not using the PCA method in our proposal. This is one of the limitations of IDSUS, which excludes decision makers from the weight measuring process (quoted in lines 68-70). The excerpt contained in lines 126-132 only details what the IDSUS weight measuring process looks like, in its original version. Our proposed weight measuring process is presented in the Materials and Methods section (lines 186-205 and Table 2).
- Materials and Method: Table 1. Please, define ICU (Intensive Care Unit)
Response: Thank you for this reminder. We made the definition.
- Materials and Method: Line 153, Please, define HDI and GDP, and RR in line 183.
Response: Thank you for this reminder. We made the definitions (Human Development Index; Gross Domestic Product; Rank Reversal - in its first appearance, line 164)
- Materials and Method: Lines 183-200 I understand that the reference 37 describe the steps of the R-TOPSIS. Then I suggest to eliminate those lines, or considering it like supplementary material (in case it is original).
Response: Thank you for the suggestion. Reference 36 presents the steps of R-TOPSIS (reference 37 presents an example of a case where R-TOPSIS proved to be robust and immune to the different RR). In any case, we have removed the indicated lines as suggested.
- Materials and Method: Lines 201-211 could be in discussion, in the broadest context possible.
Response: Thank you for the suggestion. We would like to clarify that in articles that involve MCDM, it is common for this detail to be presented in the methodology section, to facilitate the understanding of the construction of the model and to avoid the results being too long. We would also like to follow the recommendation of the procedure for solving an MCDM/A problem (Figure 1 - Almeida et al. [32]). In addition, we provide further discussion of this aspect in the new results and discussion section. Thus, to maintain a standard of construction of the approach, we would like to keep the text as it is already presented, if the reviewer agrees with this.
- Materials and Method: Line 223 and Table could be move to the “results” section.
Response: Thank you for the suggestion. As in the previous answer, we would like to keep the approach construction pattern, keeping the text and table as it is already presented, if the reviewer agrees to this.
- Results: Lines 256-269 could be in the “Discusion” section, except the sentence “the main contribution of the study is to provide a model capable of helping decision making in the healthcare area” which I see as a conclusion; and the lines 267-269 where authors express “The aim of the study…(to demonstrate its usefulness).
Response: Thank you for the suggestion. As we restructured the results and added the discussion, as per the answer to item 4, this aspect was addressed.
- Results: Lines 270-273 I see more convenient this first paragraph in “Material and methods” section, and the rest in “Results”.
Response: Thank you for the suggestion. We understand the statement, but we clarify that we follow the recommendation of the procedure for solving an MCDM/A problem (Figure 1 - Almeida et al. [32]), in which the recommendations and general observations are presented after the evaluation of the alternatives. Thus, to maintain a standard of construction of the approach, we would like to keep the text as it is already presented, if the reviewer agrees with this.
- Conclusions: I suggest to rewrite it. For example, the first sentence in line 298 is not a conclusion from my point of view. Paragraph in lines 308-312 looks like aims.
Response: Thank you for the suggestion. We rewrote the conclusions section, as per the answer to item 5.
- Conclusions: Why “R-TOPSIS method is more reliable for including new cities”? (line 301)
Response: Thank you for the observation. As stated in lines 372-374, the R-TOPSIS method is more reliable for including new cities in the analysis because there will be no risk of undesirable inversions, as it is a method immune to Rank Reversal (RR), a limitation from TOPSIS. This was explained in lines 162-181, which explain what RR is all about and how R-TOPSIS overcomes this problem. Furthermore, in lines 182-185 we reinforce the importance of an RR immune method.
- Conclusions: Finally, the last sentence could be at the end of the “Discussion” section (“may be useful”), and as future research directions.
Response: Thank you for the suggestion. We rewrote the conclusions section, as per the answer to item 5.
Reviewer 2 Report
Thank you for the opportunity to review this manuscript. The article proposes a new evaluation method the performance of the Brazilian Health System. The introduction concisely sets up the study and the literature review is sufficient. The methods are robust, and the topic will likely be of great interest to IJERPH readers around the world. Please see my detailed suggestions for improvement below:
-The results should be more detailed within the text. Currently, the authors rely on the tables to drive result interpretations. Especially given that there are no captions for the tables, results are rather difficult to discern without descriptions within the manuscript text.
-Lines 244-253: These 2 paragraphs sound more like commentary that belong in the Discussion.
-Given how interesting and innovative these results are, more attention should be given to the Discussion. Yes, the results are summarized well and supported by results. However, it would greatly improve the manuscript if the following were discussed in much more detail: a) Implications for the SUS; b) What can other countries learn from this?; and c) Highlight the innovativeness of these findings.
Author Response
Reviewer #2 Comments
Reviewer:
- Thank you for the opportunity to review this manuscript. The article proposes a new evaluation method the performance of the Brazilian Health System. The introduction concisely sets up the study and the literature review is sufficient. The methods are robust, and the topic will likely be of great interest to IJERPH readers around the world.
Response: We would like to thank the Reviewer for the positive feedback.
- The results should be more detailed within the text. Currently, the authors rely on the tables to drive result interpretations. Especially given that there are no captions for the tables, results are rather difficult to discern without descriptions within the manuscript text.
Response: Thank you for the suggestion. We agreed with the suggestion and accepted. With the changes made, we believe that the results were much better and more robust.
- Lines 244-253: These 2 paragraphs sound more like commentary that belong in the Discussion.
Response: Thank you for the suggestion. We agreed with the suggestion and complied with it, as we restructured the study, adding the discussion in the results section.
- Given how interesting and innovative these results are, more attention should be given to the Discussion. Yes, the results are summarized well and supported by results. However, it would greatly improve the manuscript if the following were discussed in much more detail: a) Implications for the SUS; b) What can other countries learn from this?; and c) Highlight the innovativeness of these findings.
Response: Thank you for the suggestion. As already stated in the answers to items 2 and 3, we improved the results and added the discussion in this section. Specifically, we believe that the points raised have been addressed in the new section.
Round 2
Reviewer 2 Report
I believe that the comments have been sufficiently addressed. I believe that this paper is suitable for publication in its current form.